# Subcellular Localizations of Catalase and Exogenously Added Fatty Acid in *Chlamydomonas reinhardtii*

**DOI:** 10.3390/cells10081940

**Published:** 2021-07-30

**Authors:** Naohiro Kato, Gabela Nelson, Kyle J. Lauersen

**Affiliations:** 1Department of Biological Sciences, Louisiana State University, Baton Rouge, LA 70803, USA; gnelso5@lsu.edu; 2Faculty of Biology, Center for Biotechnology (CeBiTec), Bielefeld University, Universitätsstrasse 27, 33615 Bielefeld, Germany; kyle.lauersen@kaust.edu.sa; 3Biological and Environmental Sciences and Engineering Division, King Abdullah University of Science and Technology (KAUST), Thuwal 23955-6900, Saudi Arabia

**Keywords:** *chlamydomonas*, microalgae, catalase, fatty acids, peroxisomes, fluorescence microscopy

## Abstract

Fatty acids are important biological components, yet the metabolism of fatty acids in microalgae is not clearly understood. Previous studies found that *Chlamydomonas reinhardtii*, the model microalga, incorporates exogenously added fatty acids but metabolizes them differently from animals and yeast. Furthermore, a recent metabolic flux analysis found that the majority of lipid turnover in *C. reinhardtii* is the recycling of acyl chains from and to membranes, rather than β -oxidation. This indicates that for the alga, the maintenance of existing acyl chains may be more valuable than their breakdown for energy. To gain cell-biological knowledge of fatty acid metabolism in *C. reinhardtii*, we conducted microscopy analysis with fluorescent probes. First, we found that CAT1 (catalase isoform 1) is in the peroxisomes while CAT2 (catalase isoform 2) is localized in the endoplasmic reticulum, indicating the alga is capable of detoxifying hydrogen peroxide that would be produced during β-oxidation in the peroxisomes. Second, we compared the localization of exogenously added FL-C16 (fluorescently labelled palmitic acid) with fluorescently marked endosomes, mitochondria, peroxisomes, lysosomes, and lipid droplets. We found that exogenously added FL-C16 are incorporated and compartmentalized via a non-endocytic route within 10 min. However, the fluorescence signals from FL-C16 did not colocalize with any marked organelles, including peroxisomes. During triacylglycerol accumulation, the fluorescence signals from FL-C16 were localized in lipid droplets. These results support the idea that membrane turnover is favored over β-oxidation in *C. reinhardtii*. The knowledge gained in these analyses would aid further studies of the fatty acid metabolism.

## 1. Introduction

The green microalga *Chlamydomonas reinhardtii* has long served as a unicellular model organism to study the regulation of carbon metabolic pathways due to its relatively simple cellular structure and evolutionary relationship to higher plants [1,2,3]. Some microalgae are known to produce considerable amounts of triacylglycerols (TAGs) under stressful environmental conditions such as nitrogen deficiency [4]. TAG is composed of individual fatty acids (FAs) in the form of acyl-chains, which are bound to a glycerol backbone. Although it is not considered an oleaginous alga, *C. reinhardtii* is used for investigations of algal lipid metabolism as it has the most well-developed molecular tools of any alga and fully sequenced as well as annotated nuclear, chloroplast, and mitochondrial genomes [5,6]. While the anabolic pathway of FAs in *C. reinhardtii* has been well studied at both molecular and systems levels [7], the breakdown and re-purposing of TAGs in algae to their component chemicals is less understood. FA turnover is an important and well-studied metabolic process in many eukaryotes including yeasts, animals and higher plants. In addition to *de novo* FA synthesis, these hydrocarbons can be liberated by TAG catabolism, wherein lipases catalyze the hydrolysis of TAGs to yield free FAs and glycerol. In animals, FAs hydrolyzed from TAGs are subject to β-oxidization in both mitochondria and peroxisomes, while in yeasts and higher plants, β-oxidization of FAs occurs only in the peroxisomes. In yeasts and higher plants, C2 acetyl-CoA units generated by β-oxidization of FAs are converted into C4 succinate units through the glyoxylate pathway for gluconeogenesis in the cytosol. As animals do not have the glyoxylate pathway, yeasts and higher plants but not animals are able to convert TAGs into carbohydrates. Enzymes belonging to both pathways have been found to be localized within peroxisomes in *C. reinhardtii* [3,8,9,10]. 

The regulation of TAG catabolic process and the fate of fatty acids (FAs) on return from nitrogen limitation remains unclear in *C. reinhardtii* and other green algae. Enzymes of both the glyoxylate cycle and β-oxidation have been localized to the peroxisomes in *C. reinhardtii* [3]. *C. reinhardtii* possess a peroxisomal acetyl-CoA synthase (ACS), an enzyme that synthesizes acetyl-CoA from acetate and CoA [9,11], which allows it to grow in the dark on exogenously added acetic acid (C2, acetate) processed by the peroxisomal glyoxylate cycle [12]. 

Despite the identification of plant-like peroxisomes, some contradictory evidence exists regarding FA catabolism as an energy generation process for *C. reinhardtii*. Since β-oxidation has been documented to occur in the peroxisomes of *C. reinhardtii* [3], a catalase is expected to be present to handle the hydrogen peroxide (H_2_O_2_) generated by this process. Catalase converts toxic H_2_O_2_ generated during β-oxidation and photorespiration to water and oxygen; however, its activity was reported to be detected from algal mitochondrial cell fractions [13]. Previous reports have argued that peroxisomal catalase should not be required for photorespiratory processes in *C. reinhardtii* (and other algae) due to the presence of a glycolate dehydrogenase in the mitochondria which does not generate H_2_O_2_ when glycolate is converted to glyoxylate [14,15]. Nevertheless, this does not rule out the need for catalase during β-oxidation. Exogenously added short-chain FAs longer than C2 acetic acid also do not increase cell density during alga mixotrophic growth [16]. Oleic acid (C18:1 long chain FA) was found not to facilitate heterotrophic growth and was even lethal to *C. reinhardtii* [3]. FA catabolism by β-oxidation should allow such heterotrophic growth on both short and long chain FAs. In addition, a recent report of meticulous isotope-labeling flux analysis demonstrated that the majority of carbon flux for FAs in this organism involves transfer of FAs from TAG to membrane lipids, but not β-oxidation. β-oxidation of FAs was found to be, at most, a modest route of TAG catabolism [17]. 

Similar to *C. reinhardtii*, the yeast *Saccharomyces cerevisiae* possesses peroxisomes in which enzymes of both the β-oxidation pathway and glyoxylate cycle are found [18]. Catalase is present in the yeast peroxisome and the organism is able to grow on exogenous FAs as an exclusive carbon source. Studies in *S. cerevisiae* revealed that exogenously added FAs are incorporated into the cells via a FA transport protein that works with long chain acyl-CoA synthetase (LACS) [19,20]. The FA transporter protein forms a functional complex with LACS in the plasma membrane. Hence, *S. cerevisiae* is able to incorporate exogenously added long-chain FAs as acyl-CoAs and grow sufficiently on these as the sole carbon source [20,21]. In *C. reinhardtii*, an orthologue of LACS is present (Cre13.g566650 and Cre17.g726700), which complements knockout mutants of *S. cerevisiae* [22]. However, no orthologue has been identified for a FA transport protein in *C. reinhardtii*. Nevertheless, we previously demonstrated that exogenous addition of fluorescently labeled palmitic acid (C16:0) to a *C. reinhardtii* culture led to FA incorporation into organelle-like cellular compartments that have bilayer membranes, termed fatty acid-induced microbodies (FAIMs) [23], indicating that exogenously added FAs are incorporated in the cells. A previous report had also found that *C. reinhardtii* predominantly metabolizes exogenously added FAs for the production of the chloroplast membranes but not in β-oxidation processes [24]. 

In this study, we sought to address the following two questions with microscopy and subcellular labelling:(1)Is there a catalase isoform localized in *C. reinhardtii* peroxisomes?(2)Where are exogenously added FAs incorporated into *C. reinhardtii?*

## 2. Materials and Methods

### 2.1. Algal Strains and Culture Conditions

*C. reinhardtii* strain CC5082 (sequence-verified wild type) were obtained from the Chlamydomonas Resource Center. *Chlamydomonas* strain UVM4 was provided by Prof. Ralph Bock [25]. The strains were maintained in 250 mL flasks containing 100 mL liquid Tris-acetate-phosphate medium (TAP) or on agar plates [26]. When the algae were cultured in nitrogen-deficient conditions, the culture was incubated in TAP-N (TAP without nitrogen) at 23 ± 2 °C under fluorescent light (60 µmol/m^2^/s) and constantly shaken on an orbital shaker at 180 rpm. 

### 2.2. Fatty Acid Preparations

For fluorescent labeling experiments, 10 mM BODIPY^®^ FL-C16 (Life Technologies) was prepared by diluting 1 mg BODIPY^®^ FL-C16 with 210 μL of DMSO. FL-C16 was added to a 3- or 4-day culture (1 to 3 × 10^6^ cells/mL) at a ratio of 200 μL to 100 mL of TAP (final concentration 20 μM) with 100 μM of oleic acid. The cultured cells were then harvested after incubation with FL-C16 as indicated time for each experiment.

### 2.3. Fluorescent Protein Constructs and Algal Transformation

All constructs were generated with the pOpt2 vector expressing fluorescent marker: mCerulean3 (pOpt2-CFP vector) or mVenus (pOpt2-YFP vector) (Wichmann et al., 2018). Targeting peptide for citrate synthase (CIS2) to express the CIS2-CFP recombinant protein was taken from our previously described plasmid constructs [9,27]. To express recombinant proteins of YFP with *Chlamydomonas* catalase 1 or 2 (CAT1 and 2, Cre09.g417150 and Cre01.g045700, respectively), the expression vectors were constructed by inserting double-stranded oligonucleotides encoding 25 amino acids of N- or C-terminal sequence of CAT1 and 2, listed in Table 1, in the pOpt2-YFP vector. The N-terminal targets were cloned into the pOpt2-YFP vector between *Nde*I-*Bam*HI (N-terminal end of YFP). The C-terminal targets were cloned into the pOpt2-YFP vector between *Eco*RV-*Eco*RI (C-terminal end of YFP without stop codon). Transformation of UVM4 was conducted by the glass-bead method as previously described [28] and colonies were selected with paromomycin (10 mg/L), zeocin (15 mg/L), or both, depending on the desired combination of constructs. Screening for the expression of each targeted marker was carried out by stereo microscopy on the agar-plate level as previously described [27] and several transformants for each construct were generated prior to confocal microscopy.

### 2.4. Fluorescence Microscopy Analysis

For all cells analyzed by fluorescence microscopy, 1 µL aliquots of cultures were dropped on a microscope slide and covered by a cover glass with a spacer. When organelle markers were included, MitoTracker^®^ Red (invitrogen), Nile Red (Sigma-Aldrich), FM1-43FX (invitrogen), or LysoTracker™ Red DND-99 (invitrogen) at final concentrations of 500 nM, 500 ng/mL, 15 μg/mL, and 150 nM, respectively, was added in the medium 30 to 90 min before the observation. The cells were viewed with a white light laser confocal microscope, Leica TCS SP8, equipped with a ×63 (N.A. 1.20) water immersion lens. Optical conditions to detect multi-fluorescent probes in a cell without bleed-through were set based on the spectra of the probes used in each experiment. The optical conditions used in each experiment are shown in Table 2. The cells were imaged at 0.65 μm z-step intervals for 30 steps. The acquired images were projected to one image using the ImageJ software (NIH, Maryland, USA, 2.0.0), Max Intensity projection type. The obtained images were manually inspected. 3-D views were created with Imaris software (Bitplane, Zürich, Switzerland, × 64 6.0.2) with volume viewer at maximum intensity projection. 

## 3. Results

### 3.1. A Catalase Isoform Is Localized in Peroxisomes 

We aimed to address the question of whether or not catalase is localized in peroxisomes of *C. reinhardtii*. Catalase activity has been shown in mitochondrial fractions of *C. reinhardtii* [13]. This report, however, was conducted prior to the characterization of algal peroxisomes as unique organelles. It cannot be ruled out that isolated mitochondria used in these studies contained peroxisomal contamination. *C. reinhardtii* has two genes annotated as catalase in its genome, Cre09.g417150 (CAT1, annotated as mono-functional catalase) and Cre01.g045700 (CAT2, annotated as catalase/peroxidase) (Phytozome v5.5). CAT1 contains a non-canonical [29], yet potential, peroxisomal targeting tripeptide type 1 (PTS1) on its C-terminus: glycine-cysteine-leucine (GCL*). Indeed, knockdown mutants of this gene had increased sensitivity to exogenous H_2_O_2_ (Michelet et al., 2013). The N- and C-terminal 25 amino acids of each isoform were added to the N- and C-terminus of the mVenus (YFP) marker, respectively, as previously described for glyoxylate cycle enzymes [9]. While the N-terminal peptides of CAT1 resulted in diffuse YFP fluorescence in the algal cytoplasm, the C-terminus resulted in clear accumulation of YFP signal in peroxisome-like organelles (Figure 1). 

All eukaryotic cells possess enzymes which use H_2_O_2_ in the endoplasmic reticulum (ER) to assist disulfide bond formation [30] and ER-localized peroxidases assist in the detoxification of this H_2_O_2_ [31]. We postulated that CAT2 was likely such a peroxidase, as both PredAlgo and signalP analysis of its amino acid sequence predicts a secretory signal on its N-terminus [32,33]. Accordingly, the N-terminus of CAT2 was found to result in YFP marker secretion from *C. reinhardtii* (Figure 1, stereo) and inclusion of the C-terminal portion of CAT2 to this construct resulted in retention of the protein in the endoplasmic reticulum (Figure 1). The terminal 4 amino acids of CAT2 are phenylalanine-valine-glutamate-leucine (FVEL) which seem to function in *C. reinhardtii* similar to the canonical endoplasmic reticulum retention signal KDEL [34].

### 3.2. Exogenously Added Fatty Acids Are Incorporated into Cells within 10 Min via a Non-Endocytic Mechanism and Are Not Directed to Lysosomes

We then aimed to determine how exogenously added FAs are incorporated into the cells in *C. reinhardtii*. To this end, the cells were imaged by confocal fluorescence microscopy after the addition of BODIPY-conjugated fluorescent palmitic acid (C16:0) to the culture medium (hereafter, FL-C16). BODIPY-conjugated FAs have been used to study FA uptake and trafficking in multiple organisms [35,36]. They are readily metabolized to phospholipids, di- and triacylglycerols, and other lipids in living cells. In animal cells, incorporated BODIPY-conjugated FAs can be identified in the plasma membrane and organelles by fluorescence microscopy. We previously found that when FL-C16s were added to *C. reinhardtii* culture, it was incorporated into compartments containing a lipid bilayer which were termed fatty acid-induced microbodies (FAIMs) [23]. 

To investigate the mechanism of FA incorporation in *C. reinhardtii,* we first monitored FL-C16 incorporation by time-lapse microscopy. This analysis revealed FL-C16 speckles in the cytosol within 10 min after addition to the medium (Figure 2A). A green lipophilic fluorescent dye (FM1-43) was added to the medium in addition to yellow FL-C16 as it is incorporated into the cells via the endocytic mechanism and stains the membranes [37]. During endocytosis, extracellular compounds are first surrounded by the plasma membrane and then the membrane is pinched off to form vesicles containing endocytosed compounds. FM1-43 marks the plasma membrane, endosomes, vesicles, and vacuolar membranes. Here, the signals of FL-C16 do not co-localize with those of FM1-43 (Figure 2B), strongly suggesting that *C. reinhardtii* does not incorporate exogenously added FA via the endocytic mechanism. Compartmentalization of exogenously added FA may occur via the autophagy-lysosome pathway which digests overloaded FAs within cells [38]. LysoTracker was also added to the medium in addition to 20 μM FL-C16. The fluorophore of LysoTracker is activated only at low pH and selectively stains acidic organelles such as lysosomes and autophagosomes [39]. Here, LysoTracker signals did not co-localize with those of FL-C16 (Figure 2C), indicating that FL-C16 is not in the acidic regions in the cells, at least during incorporation. 

### 3.3. FAIMs Are Not Peroxisomes or Sub-Compartments in Mitochondria

We then addressed the question where exogenously added FAs are transported in *C. reinhardtii*. Peroxisomes seemed to be the most reasonable subcellular compartment if exogenously added FAs would undergo β-oxidation. We postulated that previously observed FAIMs may have been peroxisomes as they showed similar patterns as fluorescently labeled peroxisomal proteins in microscopy [9,23]. An alternative compartment for FA accumulation could be mitochondria because this organelle may also be involved in β-oxidation in some organisms [3,40,41]. Previously, we determined that enzymes of the glyoxylate cycle in *C. reinhardtii* were localized in the peroxisomes using of the N- or C-termini of these enzymes fused to mVenus yellow fluorescence protein (YFP) [9]. We used the N-terminal 25 amino acids of citrate synthase (CIS2) to test FAIM-peroxisome co-localization as a representative glyoxylate cycle enzyme. Fusion to CFP permitted clear separation from FL-C16 yellow fluorescence as previously described for the co-localization of GPR1/FUN34/YaaH (GFY) transporter proteins [42]. Frequent co-localization of FL-C16 yellow fluorescence with that of CIS2-CFP was expected if FAIMs and peroxisomes were the same organelle. Alternatively, signals would co-localize with MitoTracker (mitochondrial dye) if this was their subcellular destination. Surprisingly to us, multicolor confocal microscopy of the cells with all three markers revealed that FL-C16 signals do not colocalize with CIS2-CFP or MitoTracker (Figure 3), indicating that FAIMs are independent of peroxisomes and mitochondria. We confirmed that signal separation of FL-C16 and CIS2-CFP is not due to compartment movement during the acquisition of z-stacked images in the live cells by constructing a 3-D view from stacked images (Appendix A). We found that the compartments do not move significantly during image acquisition over ~30 s. These findings indicate that the majority of exogenously added and incorporated FAs are not transported into peroxisomes in *C. reinhardtii*, although we cannot rule out that there may be some below the limit of fluorescence microscopy detection.

### 3.4. Exogenously Added FA Are Localized in Lipid Droplets

It is known that intracellularly generated long chain FAs are incorporated into TAGs when *C. reinhardtii* is cultured in nitrogen-deficient conditions [17,43]. To examine the fate of FL-C16 during cellular TAG formation, we cultured *C. reinhardtii* expressing CIS2-CFP in a medium with FL-C16 but without nitrogen (TAP-N) for 4 d. Lipid droplets were visualized by the neutral lipid staining dye Nile red [23]. Multicolor confocal microscopy of the cells revealed that FL-C16 signal co-localized with Nile red signal but not CIS2-CFP under nitrogen-replete or -deficient conditions (Figure 4). CIS2-CFP fluorescence intensities were reduced in TAP-N, owing to lower overall protein turnover in the starved state, but were nevertheless detectible. 

## 4. Discussion

### 4.1. Peroxisomal Localization of CAT1 Supports the Idea That the β-Oxidation Occurs in the Peroxisome 

We previously determined that peroxisomes exist in *C. reinhardtii* and that most of the glyoxylate cycle is localized there, including an acetyl-CoA synthase (ACS3) [9]. This localization was observed with both peroxisome targeting peptide types (PTS1 and PTS2) which use different import machinery. It was also previously observed that the key enzyme involved in β-oxidation, acyl-CoA oxidase 2 (ACX2), was co-localized with the glyoxylate-cycle associated MDH2 in the peroxisomes [3] and recombinant ACX2 catalyzes the oxidation of fatty acyl-CoA into trans-2-enoyl-CoA and produces H_2_O_2_ in vitro. Accordingly, it is highly likely that functional β-oxidation occurs within the peroxisomes in *C. reinhardtii*, especially during turnover processes of endogenous lipids and membranes. In this study, we determined that catalase isoform 1 (CAT1) contains a non-canonical PTS1 and is localized in peroxisomes (Figure 1). This result is important for several reasons: it indicates that the alga has alternative PTS motifs than those of higher plants, further complicating bioinformatic prediction of PTS in alga; previously, catalase activity had been associated with the algal mitochondria and alternative pathways suggested for non-H_2_O_2_ inducing processes to circumvent the need for its activity [14,15]; the presence of a catalase in algal peroxisomes suggests that these organelles should be capable of conducting β-oxidation as recently suggested [3]. 

### 4.2. Separation of FAIMs and Peroxisomes Supports Membrane Turnover Observations

Comparative fluorescence staining demonstrated that FAIMs where exogenously added FAs are accumulated are separate compartments from the peroxisomes (Figure 3). We found that the FAIMs are not artefacts within mitochondria and lysosomes (Figure 2 and Figure 3). One possibility for these localizations may be that the fluorescent dye, BODIPY, is too large to pass through certain FA transporters. However, this is unlikely because BODIPY conjugated FAs have been previously shown to localize in the organelles in other organisms. They are rapidly metabolized to membranes and transported to peroxisomes [44] and mitochondria [45]. Our results suggest that FAIMs are non-peroxisomal compartments that accumulate exogenously added FAs in *C. reinhardtii* (Figure 4). It has been recently demonstrated by flux analysis that the utilization of FAs by b-oxidation is at most a modest route of FA catabolism during TAG lipolysis; rather, the majority of FAs are directly channeled to membranes in both endoplasmic reticulum and chloroplasts [17]. A previous study in *C. reinhardtii* using ^14^C-lableled palmitic and oleic acids found that these are mainly used to produce betaine lipids for the plastid membrane [24]. Our microscopy analyses support findings of these previous studies.

### 4.3. Biogenesis of FAIMs Remains a Question

We found that FL-C16 is incorporated in lipid droplets in nitrogen-deficient conditions (Figure 4) similar to observations in specific types of animal cells [45,46,47,48]. Previous studies in yeasts and animals found fluorescently labeled FA specifically accumulate in the lipid droplets of the cells when the metabolic state is directed towards TAG synthesis [45,46,47,48,49]. An example is that external oleic acid addition to *S. cerevisiae* stimulates the metabolic switch to lipid droplet formation [50]. When yeast or animal cells are not in a state of TAG accumulation, fluorescently labeled FAs are observed as diffuse signals in the cytoplasm [20,36,51,52,53]. The unique aspect of *C. reinhardtii* is that FAIMs are formed even in nitrogen-replete conditions, suggesting that the upregulation of TAG biosynthesis is not required for exogenous FAs to be transported into this subcellular compartment (Figure 2, Figure 3 and Figure 4). We found signals of Nile red and FL-C16 are co-localized in both nitrogen-deficient and -replete conditions, even though the signals of Nile red are extremely low in the nitrogen-replete condition (Figure 4 and Appendix A). We were not able to determine whether the weak signal of Nile red in N-replete conditions is due to low concentration of TAGs or simply hydrophobic environment in the lumen of FAIMs. Nonetheless, FAIMs accumulate hydrophobic FL-C16 metabolites that may include low amounts of TAGs. We previously found with transmission electron microscopy that FAIMs have a bilayer [23]. In *S. cerevisiae*, biogenesis of lipid droplets and peroxisomes occur at the same domains in the ER and de novo synthesized lipid droplets and pre-peroxisome vesicles often remain associated with each other [50]. Further proteomic analyses on isolated FAIMs and peroxisomes would address the question, something which the separate localization of PTS-tagged fluorescent proteins and FL-C16 may help facilitate.

## Figures and Tables

**Figure 1 cells-10-01940-f001:**
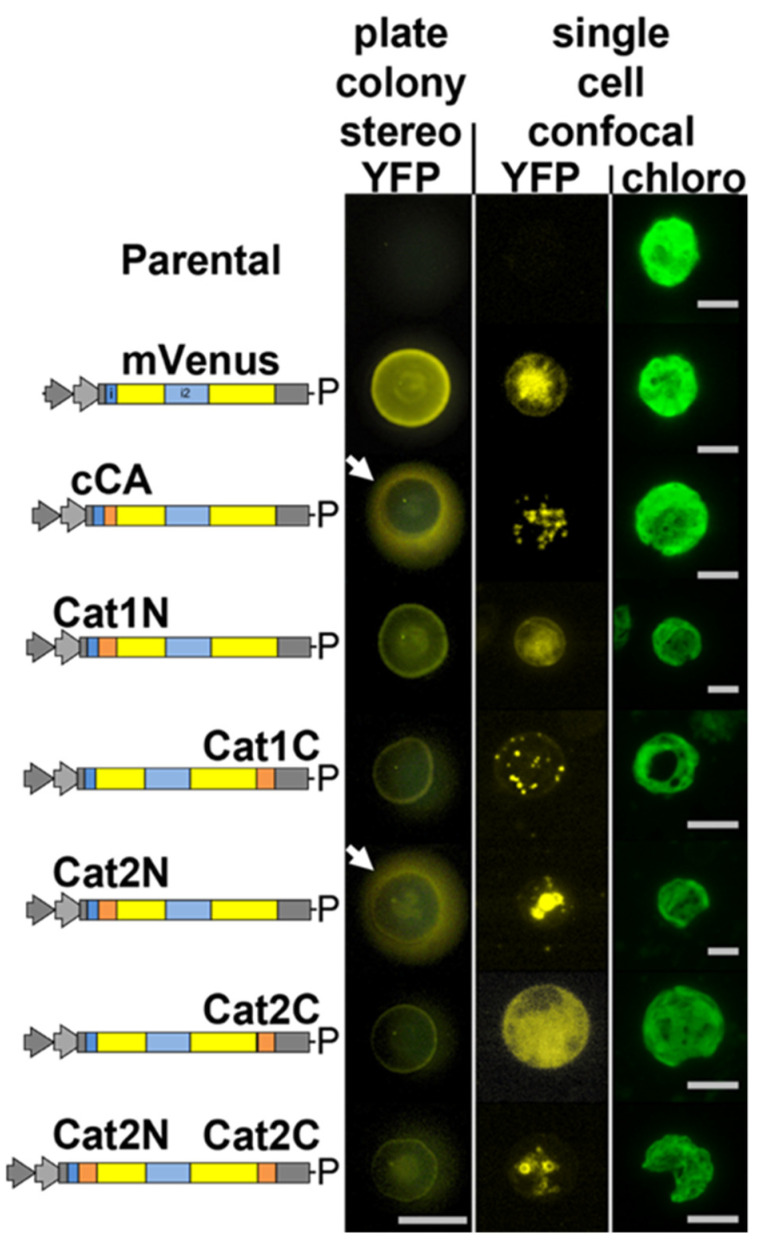
Catalase 1 (Cat1, Cre09.g417150) is localized in peroxisomal microbodies in *C. reinhardtii* mediated by a non-canonical PTS1 signal. The genome of *C. reinhardtii* contains two catalase isoforms: catalase 1 contains the terminal tripeptide GCL*, which is a non-canonical PTS1. Here, the 25 amino acids from the N- and C-termini of each of these enzymes (orange boxes) were added to the mVenus (YFP) marker in the pOpt2_YFP vector (Lauersen et al., 2015) and expressed in *C. reinhardtii*. Vector diagrams correspond to the genetic elements of the pOpt2 vectors, YFP is shown in yellow and contains an intron (blue box) as previously reported (Lauersen et al., 2015), P-paromomycin resistance cassette. Colonies were analyzed by stereo- and confocal fluorescence microscopy, individual colonies on an agar plate (stereo) or single cells (confocal) are shown. Chlorophyll fluorescence is shown for cell orientation in confocal analyses. The C-terminus of Cat1-mediated YFP localization into peroxisomal microbodies, while the N-terminus of Cat2 targeted the marker for secretion similar to the previously characterized *C. reinhardtii* carbonic anhydrase 1 (cCA) secretion signal (white arrows). Addition of its C-terminus to the Cat2 N-terminal construct resulted in ER retainment of the marker. Thirty optical sections of a 20 μm z-stack are projected into one image with their maximum intensity. Stereo scale bar represents 1 cm, Confocal scale bars represent 5 µm.

**Figure 2 cells-10-01940-f002:**
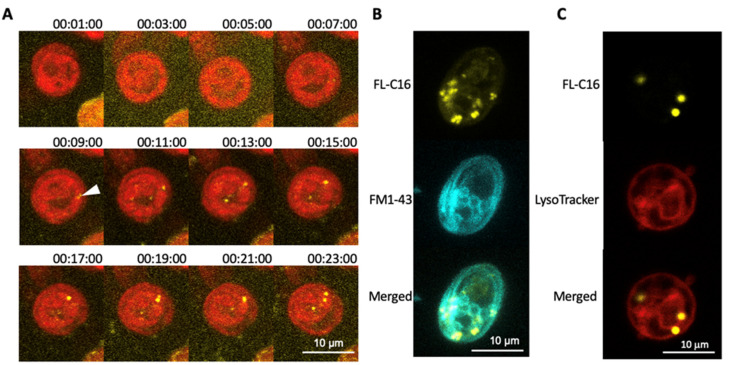
Exogenously added fatty acids are compartmentalized within 10 min via non-endocytic mechanism. (**A**) *C. reinhardtii* UVM4 was observed with confocal laser scanning microscopy every 2 min immediately after adding 20 μM fluorescently labelled palmitic acid (FL-C16) in TAP medium. Thirty optical sections of a 20 µm z-stack with a FL-C16 fluorescence channel (yellow) and chlorophyll fluorescence channel (red) are projected into one image with their maximum intensity, respectively. Numbers above each image show time (h:min:sec) after adding FL-C16. Scale bars represent 10 µm. Notice FL-C16 fluorescence signals are clearly detectable at 9 min after adding FL-C16 (arrowhead). (**B**) *C. reinhardtii* CC5082 was observed with confocal laser scanning microscopy 1 h after adding 20 µm FL-C16 and 10 µM FM1-43. Thirty optical sections of a 20 µm z-stack with a FL-C16 fluorescence channel (yellow) and FM1-43 fluorescence channel (cyan) are projected into one image with their maximum intensity. Scale bars represent 10 µm. FL-C16 fluorescence signals are not detected in the plasma membranes nor endosomes. (**C**)*. reinhardtii* UVM4 was observed with confocal laser scanning microscopy 1 h after adding 20 µm FL-C16 and 10 µM LysoTracker. Thirty optical sections of a 20 µm z-stack with a FL-C16 fluorescence channel (yellow), LysoTracker fluorescence channel (red) are projected into one image with their maximum intensity, respectively. Transmission image is also shown. Scale bars represent 10 µm. FL-C16 fluorescence signals are not co-localized with LysoTracker signals. Scale bars represent 10 µm.

**Figure 3 cells-10-01940-f003:**
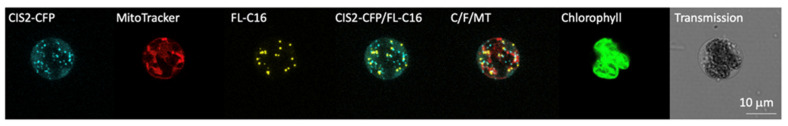
Exogenously added fatty acids are not accumulated in the peroxisomes or mitochondria. *C. reinhardtii* UVM4 expressing CIS2-CFP (cyan) was observed with confocal laser scanning microscopy after adding 20 µM FL-C16 (yellow), and 10 µM MitoTracker (red). Thirty optical sections of a 20 µm z-stack for each signal are projected into one image with their maximum intensity, respectively. Independent fluorescence channels are shown for CIS2-CFP, FL-C16, and MitoTracker as well as an overlay of these signals (C/F/MT). Chlorophyll fluorescence and transmission image are also shown. CIS2 is the citrate synthase, a key enzyme characterized to localize in algal peroxisomes where it participates in the glyoxylate cycle. Neither the signals of CIS2-CFP or MitoTracker co-localize with FL-C16. Chlorophyll autofluorescence is shown in green.

**Figure 4 cells-10-01940-f004:**
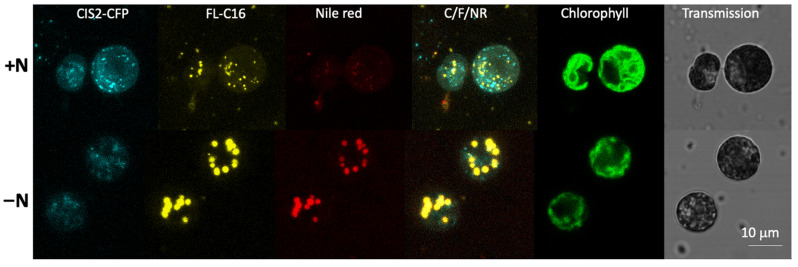
Exogenously added FAs are found in lipid droplets in nitrogen-deficient conditions. *C. reinhardtii* UVM4 expressing CIS2-CFP was cultured in nitrogen-replete (+N) or -deficient (−N) conditions. FL-C16 (20 µM) was added to the culture medium 3 d before or 1 h before microscopy for the +N condition or −N condition, respectively. FL-C16 signals (yellow) were not co-localized with peroxisomal CIS2-CFP signals (cyan) in either condition. Nile red signals co-localized with FL-C16 in the −N condition.

**Table 1 cells-10-01940-t001:** Oligonucleotides used to construct targeting peptides of CAT1 and CAT2.

Target	Oligonucleotide 5′–3′
CAT1-NFw	AATTTCATATGGACCCCGCCAAGATCCGCCCCAGCAGCGCCTACAACACCCCCTACTG
CAT1-NRw	AAATTGGATCCCACGGGGGCGCCGCTGTTGGTGGTCCAGTAGGGGGTGTTGTAGGCGCTGC
CAT1-CFw	AATTTGATATCGTGGGCTACTGGAGCCAGGCCGACCCCCAGCTGGGCGCCCGCATCGCCGC
CAT1-CRv	AAATTGAATTCTTACAGGCAGCCGCGGCCCTGCAGCTTGGCGGCGATGCGGGCGCCCAGCTGGG
CAT2-NFw	AATTTCATATGCGCGACAAGGCCCTGATCACCCTGCTGCTGGCCGCCAGCGCCGCCTT
CAT2-NRv	AAATTGGATCCGCCGAAGGGGCACTTGCAGGTGGCGAAGGCGGCGCTGGCGGCCAGCAGCA
CAT2-CFw	AATTTGATATCGCCGCCGTGGTGCAGCTGGTGCTGGAGGCCAGCTACGGCGCCGCCGCCCG
CAT2-CRv	AAATTGAATTCTTACAGCTCCACGAACTCCTCCTCCACGCGGGCGGCGGCGCCGTAGCTGGCCT

**Table 2 cells-10-01940-t002:** Optical conditions to resolve spectra of probes individually.

**Set 1**	**Excitation (nm)**	**Emission (nm)**
CFP	458	463–490
FLC-16	500	505–530
MitoTracker	580	585–620 *
Chlorophylls	458	697–800
**Set 2**	**Excitation (nm)**	**Emission (nm)**
CFP	458	463–490
FLC-16	500	505–530
Nile Red	540	545–600 *
Chlorophylls	458	697–800
**Set 3**	**Excitation (nm)**	**Emission (nm)**
FL-C16	502	507–540
FM1-43	470	565–655
**Set 4**	**Excitation (nm)**	**Emission (nm)**
FL-C16	500	505–532
LysoTracker	576	583–643C

* 0–6.0, ** 0.4–7.5, *** 0.2–7.5 (lightgate time in nanoseconds).

## Data Availability

Not applicable.

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
