# Peer review of "Subcellular Localizations of Catalase and Exogenously Added Fatty Acid in Chlamydomonas reinhardtii"

_cells, 2021, doi:10.3390/cells10081940_

Round 1

Reviewer 1 Report

This paper deals with study on lipids metabolism in Chlamydomonas reinhardtii, using fluorescence microscopy to explore the subcellular localizations of catalase and the incorporation of exogenous fatty acids.

The study is well conducted, with results supporting conclusions of authors. This is a good study, giving new insight into microalgae's lipids metabolism.

Only minor comments:

-p2, lines 49-52 and lines 59-63: these 2 parts are a little bit redundant, probably it could be merged.

-p2, lines 79-86: font is not the same

-p3, lines 112-113: 2 Chlamydomonas strains are mentioned in the material and methods section: CC5082 and UVM4. The UVM4 was used for transformations but it is not clear for me for CC5082; which experiments were conducted with?

-p3, lines 118-120: "When the algae were cultured autotrophically, the culture was incubated in minimal medium (MM, TAP without acetate) at 23 ± 2°C in the dark and constantly shaken on an orbital shaker at 180 rpm." cultured autotrophically but in the dark?

Author Response

We appreciate valuable comments.

Below is  our point-by-point response to the reviewer’s comments:

POINT: -p2, lines 49-52 and lines 59-63: these 2 parts are a little bit redundant, probably it could be merged.

RESPONSE: We agreed with the reviewer that the two parts were somewhat redundant. We compiled the two parts into one:

In animals, FAs hydrolyzed from TAGs are subject to β-oxidization in both mitochondria and peroxisomes, while in yeast and higher plants, β-oxidization of FAs occurs only in the peroxisomes. In yeasts and higher plants, C2 acetyl-CoA units generated by β-oxidization of FAs are converted into C4 succinate units through the glyoxylate cycle for gluconeogenesis in the cytosol. Because animals do not have the glyoxylate pathway, yeasts and higher plants but not animals are able to convert TAGs into carbohydrates. Enzymes belonging to both pathways have been found localized within peroxisomes in C. reinhardtii (Cooper and Beevers, 1969; Pracharoenwattana and Smith, 2008; Lauersen et al., 2016; Kong et al., 2017).”

in p2 line 49-57 in the revised manuscript.

POINT: -p2, lines 79-86: font is not the same

RESPONSE: We corrected the font so that the font of the part is same as other parts.

POINT: -p3, lines 112-113: 2 Chlamydomonas strains are mentioned in the material and methods section: CC5082 and UVM4. The UVM4 was used for transformations but it is not clear for me for CC5082; which experiments were conducted with?

RESPONSE: CC5082 (wild type Chlamydomonas used in the genome project) was used to examine colocalization of FL-C16 and membranes (FM1-43 staining, Fig. 2B). The usage was described in the figure legend in Fig.2 in the previously submitted manuscript. While CC5082 possesses cell wall, UVM4 does not. We wanted to confirm that FL-C16 is incorporated within cells independent from the status of cell wall. Usage of UVM4 for the time course experiment (Fig. 2A) and colocalization analysis of FL-C16 with lysosomes (LysoTracker staining, Fig. 2C)  was also described in the figure legend in Fig.2 in a similar manner.

POINT: -p3, lines 118-120: "When the algae were cultured autotrophically, the culture was incubated in minimal medium (MM, TAP without acetate) at 23 ± 2°C in the dark and constantly shaken on an orbital shaker at 180 rpm." cultured autotrophically but in the dark?

RESPONSE: The sentence should have removed in the previously submitted manuscript because we did not present any data of autotrophic growth or cultures in the dark. The sentence is scientifically incorrect anyway. We removed the sentence in the revised manuscript.

Reviewer 2 Report

Dear Authors, 

The work presented for evaluation includes basic research in the field of cell biology. I value the results of this work highly. The presentation of the results is correct and transparent. Below are some minor fixes and comments. Line 50-51: "In yeasts and higher plants, this is followed by conversion of acetyl-CoA into C4 carbon units by the glyoxylate cycle for gluconeogenesis." Gluconeogenesis is not a cycle. In biochemistry, the term pathway is adequate for gluconeogenesis. Please edit this sentence to make the tone clear. Line 79-86: Please correct the font Table S1 I recommend that you move to the main manuscript. The sequences of the oligonucleotide probes used are important to researchers and should not be hidden in the supplement. After taking into account minor corrections, it recommends accepting it for publication. 

Author Response

We appreciate valuable comments.

Below is  our point-by-point response to the reviewer’s comments:

POINT: Line 50-51: "In yeasts and higher plants, this is followed by conversion of acetyl-CoA into C4 carbon units by the glyoxylate cycle for gluconeogenesis. "Gluconeogenesis is not a cycle. In biochemistry, the term pathway is adequate for gluconeogenesis. Please edit this sentence to make the tone clear. 

RESPONSE: To clarification, we modified the sentences to:

In yeasts and higher plants, C2 acetyl-CoA units generated by b-oxidization of FAs are converted into C4 succinate units through the glyoxylate pathway for gluconeogenesis in the cytosol.”

in Line 51-53 in the revised manuscript.

POINT: Line 79-86: Please correct the font Table S1 I recommend that you move to the main manuscript. The sequences of the oligonucleotide probes used are important to researchers and should not be hidden in the supplement. 

RESPONSE: We moved Table S1 to the main manuscript as Table 1. We also moved Table S2 (filter setting for fluorescence detection) to the main manuscript as  Table 2 because we believe the optical condition in fluorescence microscopy is as important as oligo sequence. Table 1 and 2 are appeared in Line 142 and 158, respectively, in the revised manuscript.

Reviewer 3 Report

The manuscript analyzes the localization of two fluorescent protein fused catalases present in the genome of green alga Chlamydomonas reinhardtii. Furthermore, it follows the internalization and localization of exogenously added fluorescently labeled palmitic acid. The localization analyses suggest that one catalase, CAT1 is peroxisome localized, while the other one, CAT2, is localized to endoplasmic reticulum. The results of internalization of fluorescently labeled palmitic acid suggest that it is internalized into the algal cells within about 10 minutes and it localizes to specific structures distinct from both peroxisomes and mitochondria. A co-localization experiment indicate that the fatty acid may be internalized into lipid bodies. The work consists primarily of confocal microscopy data. The results have interesting implication that are extensively discussed both with the Results and Discussion sections. The results are interesting both from the basic research point of view and have potential biotechnological application.

I have several comments and suggestions:

  1. 118-119 It does not make sense how the algae were to grow autotrophically in dark?
  2. 125 What was the volume/cell concentration harvested for the analyses?
  3. 128-142 This paragraph seems logically off as the description starts with the last step in the process and ends with the first step. It is not clear what the targeting peptide was used for and where it was cloned. Also it is not clear whether C- or N-terminal fusions with fluorescent proteins were generated.
  4. 145-148 This sentence seems to miss a verb, please re-phrase.
  5. 150-151 This is unclear, please, re-phrase.
  6. 169-170 Please specify in the text if the amino acids were N- or C-terminal to YFP?
  7. 173-174 „This supports the concept that peroxisomal processes involve H2O2…“ Do you mean peroxisomal metabolism?

Figure 1              Is the localization to peroxisomes or ER based simply on the morphology of the organelles or were there co-localization experiments with organelle markers done? The co-localization with markers should be provided to strengthen the outcome.

  1. 213-216 “We previously found that…” This has been already stated in Introduction, no need to duplicate the information.
  2. 327-328 Could the authors speculate how the fluorescently labeled FA entered the cells in the apparent absence of FA transporter in the alga?

Author Response

We appreciate valuable comments.

Below is  our point-by-point response to the reviewer’s comments:

  1. 118-119 It does not make sense how the algae were to grow autotrophically in dark?

RESPONSE: The sentence should have removed in the previously submitted manuscript because we did not present any data of autotrophic growth or cultures in the dark. The sentence is scientifically incorrect anyway.  We removed the sentence in the revised manuscript.

  1. 125 What was the volume/cell concentration harvested for the analyses?

 RESPONSE: It was 1 to 3 x 106  cell/ mL. For clarification, we modified the section to:

FL-C16 was added to a 3- or 4-day culture (1 to 3 x 106 cells/mL) at a ratio of 200 μL to 100 mL of TAP……...”

in Line 120 in the revised manuscript.

  1. 128-142 This paragraph seems logically off as the description starts with the last step in the process and ends with the first step. It is not clear what the targeting peptide was used for and where it was cloned. Also it is not clear whether C- or N-terminal fusions with fluorescent proteins were generated.

RESPONSE: To clarify the process and fusion side of the fluorescent protein, we modified the entire section to:

All constructs were generated with the pOpt2 vector expressing fluorescent marker: mCerulean3 (pOpt2-CFP vector) or mVenus (pOpt2-YFP vector) (Wichmann et al., 2018). Targeting peptide for citrate synthase (CIS2) to express the CIS2-CFP recombinant protein was taken from our previously described plasmid constructs (Lauersen et al., 2015; Lauersen et al., 2016). To express recombinant proteins of YFP with Chlamydomonas catalase 1 or 2 (CAT1 and 2, Cre09.g417150 and Cre01.g045700, respectively), the expression vectors were constructed by inserting double-stranded oligonucleotides encoding 25 amino acids of N- or C-terminal sequence of CAT1 and 2, listed in Table 1, in the pOpt2-YFP vector. The N-terminal targets were cloned into the pOpt2-YFP vector between NdeI-BamHI (N-terminal end of YFP). The C-terminal targets were cloned into the pOpt2-YFP vector between EcoRV-EcoRI (C-terminal end of YFP without stop codon). Transformation of UVM4 was conducted by glass-bead method as previously described (Kindle, 1990) and colonies were selected with paromomycin (10 mg/L), zeocin (15 mg/ L), or both depending on the desired combination of constructs. Screening for expression of each targeted marker was done by stereo microscopy on the agar-plate level as previously described (Lauersen et al., 2015) and several transformants for each con-struct were generated prior to confocal microscopy.”

in Line 124-140 in the revised manuscript. Following the Reviewer 2 suggestion, we moved Table S1 that shows sequences of the oligonucleotides in CAT1 and 2 targeting peptides were moved to the main text as Table 1 (Line 142) in the revised manuscript.

  1. 145-148 This sentence seems to miss a verb, please re-phrase.

RESPONSE: We corrected the grammar error. Namely, the section was rephrased to:

When organelle markers included, MitoTracker® Red (invitrogen), Nile Red (Sig-ma-Aldrich), FM1-43FX (invitrogen), or LysoTracker™ Red DND-99 (invitrogen) at final concentrations of 500 nM, 500 ng / mL, 15 mg / mL, and 150 nM, respectively, was added in the medium 30 to 90 min before the observation.”

in Line 146-150 in the revised manuscript.

  1. 150-151 This is unclear, please, re-phrase.

RESPONSE: We rephrased to:

Optical conditions to detect multi fluorescent-probes in a cell without blead-through were set based on the spectra of the probes used in each experiment. The optical conditions used in each experiment are shown in Table 2”.

in Line 151-154. Table S2 that shows the optical conditions in the previous manuscript was moved to the main text as Table 2 in Line 159 in the revised manuscript.

  1. 169-170 Please specify in the text if the amino acids were N- or C-terminal to YFP?

RESPONSE:  To clarify the fusion end of the targeting peptide, we modified the sentence to:

The N- and C-terminal 25 amino acids of each isoform were added to the N- and C-terminus of the mVenus (YFP) marker, respectively

in Line 173 – 174 in the revised manuscript.

  1. 173-174 „This supports the concept that peroxisomal processes involve H2O2…“ Do you mean peroxisomal metabolism?

RESPONSE:  We noticed the part was redundantly discussed in the discussion section (Line 314-321). Accordingly, we omitted Line 173-174 .  The discussion part was not alternated in the revised manuscript.

Figure 1              Is the localization to peroxisomes or ER based simply on the morphology of the organelles or were there co-localization experiments with organelle markers done? The co-localization with markers should be provided to strengthen the outcome.

RESPONSE:  We were not able to obtain  ER markers to strength the outcome.  To address the question differently, we provided two independent  references that computationally predict the ER localization of CAT2 (Line 179 – 181 in the revised manuscript).  We also provided our own data that support the computational prediction that is secretion and inclusion of CAT2 by the N- and C-terminal targeting peptide, respectively (Fig. 1). We believe these would suffice (even if it’s not strongest) to present CAT2 as an ER localized catalase. Most importantly our data provides the evidence that CAT1 and CAT2 localize differently.

  1. 213-216 “We previously found that…” This has been already stated in Introduction, no need to duplicate the information. 

RESPONSE:  We removed the sentence in the revised manuscript.

  1. 327-328 Could the authors speculate how the fluorescently labeled FA entered the cells in the apparent absence of FA transporter in the alga?

 RESPONSE:  We speculate Chlamydomonas would possess FA transporters in the plasma membrane even though homologous genes from yeast are not identified in the genome. Chlamydomonas does possess FA transporters, which are not found in yeast,  such as one for FA transport between the chloroplasts and ER. We speculate that these FA transporters may be localized in the plasma membrane as well.  We did not describe the speculation in the manuscript because our speculation is too naïve to publish at this point and not the main focus of this manuscript. 

Round 2

Reviewer 3 Report

The comments and suggestions were answered accordingly. I have no further comments.